# Male Breast Cancer: From Molecular Genetics to Clinical Management

**DOI:** 10.3390/cancers14082006

**Published:** 2022-04-15

**Authors:** Matilde Pensabene, Claudia Von Arx, Michelino De Laurentiis

**Affiliations:** National Cancer Institute, IRCCS Fondazione G. Pascale, 80131 Naples, Italy; claudia.vonarx@istitutotumori.na.it (C.V.A.); m.delaurentiis@istitutotumori.na.it (M.D.L.)

**Keywords:** male breast cancer, BRCA1/2, PALB2, CHECK2, cancer genetic counseling, genetic test, prevention, hereditary cancer syndromes

## Abstract

**Simple Summary:**

Male breast cancer (MBC) is a rare disease. Genetic factors predispose to male breast cancer. Germline and/or genetic and/or epigenetic alterations at the somatic level identify a subset of male breast cancer that could differ from female breast cancer (FBC). Cancer genetic counseling should be included in the work-up of male breast cancer to identify the possible genetic origin of the tumor and to offer patients and their at-risk family members adequate management.

**Abstract:**

MBC is a rare disease accounting for almost 1% of all cancers in men and less than 1% of breast cancer. Emerging data on the genetic drivers of predisposition for MBC are available and different risk factors have been associated with its pathogenesis. Genetic alterations, such as pathogenetic variants in BRCA1/2 and other moderate-/low-penetrance genes, along with non-genetic risk factors, have been recognized as pathogenic factors for MBC. Preventive and therapeutic implications could be related to the detection of alterations in predisposing genes, especially BRCA1/2, and to the identification of oncogenic drivers different from FBC. However, approved treatments for MBC remain the same as FBC. Cancer genetic counseling has to be considered in the diagnostic work-up of MBC with or without positive oncological family history. Here, we review the literature, reporting recent data about this malignancy with a specific focus on epidemiology, and genetic and non-genetic risk factors. We introduce the perspective of cancer genetic counseling for MBC patients and their healthy at-risk family members, with a focus on different hereditary cancer syndromes.

## 1. Introduction

MBC is a rare disease [1]. Current evidence about treatment is derived from small single institutional experiences. The knowledge of genetic drivers of MBC could drive prospective clinical trials of more specific and targeted therapies.

Here, we review the literature, reporting recent data about this malignancy, with a specific focus on epidemiology and predisposing genes. We highlight the differences between MBC and FBC, especially in terms of oncogenic drivers, differences that could guide future research in individualized MBC treatment. This review also focuses on hereditary cancer syndromes and cancer genetic counseling, including prevention, which represent essential aspects of MBC management.

## 2. Methods

A systematic literature search was conducted for articles in English on MBC and BRCA genes, using the PubMed database, with no limitation on the publication date or journal. Searches were made using the terms ‘male breast cancer’ and ‘BRCA’ and ‘predisposing genes’. Selected articles were published between 2000 and 2021. Given the low frequency of MBC and the lack of prospective clinical trials, we only performed a descriptive analysis.

## 3. Epidemiology

MBC represents less than 1% of all cancers in males and less than 1% of breast cancer (BC) overall [1]. Recently, the International Agency for Research on Cancer (IARC) provided an international comparison of MBC and FBC, deriving data from 88 registries of various ethnic groups in America, Europe, Asia, and Oceania, a markedly higher MBC incidence was reported in North America compared to South and Central America, similarly with FBC. In Europe, Italy registered the highest incidence of both tumors. Moreover, the lowest incidence rates of FBC and MBC were registered in Thailand for age while the highest overall age-adjusted incidence were registered in Israel (lifetime risk is about 1 in 1190 men per year) [2].

Data from the registry of Surveillance, Epidemiology, and End Results (SEER), which collected a total of 5494 MBC cases diagnosed between 1973 and 2005, offer better insight into MBC’s characteristics. According to this report, 11% of cases were diagnosis of in situ carcinoma. At diagnosis the median age was 67 years vs 61 years in females, and advanced stage at diagnosis was more frequent in men than in female, with a mean tumor size of 2.4 cm and lymph node involvement was more common in men than in women. MBC appeared to be more likely to be hormone driven, with only 7.6% of breast cancers being negative for estrogen receptor (ER) in men compared to 23% in women [3].

Miao et al. compared relative survival outcomes of 2665 men and 459,846 women with BC. They reported that men have a poorer 5-year relative survival compared to women, with a relative excess risk (RER) of 1.27. However, when these data are adjusted for age, year of diagnosis, stage, and treatment the relative survival for BC of male is longer than the one of female [4]. A single-center retrospective analysis of 47 patients showed poorer long-term survival in MBC carriers of pathogenetic variants of the BRCA1/2 genes compared to wild-type subjects [5]. Differences in the prognosis between FBC and MBC could be related to different genetic drivers other than BRCA genes, as recently shown [6,7,8].

## 4. Risk Factors

### 4.1. Genetic Risk Factors

A high risk of MBC was reported in men with an affected first-degree family member (relative risk (RR) = 1.92, 95% CI: 1.19–3.09) [9]. Recently, Lecarpentier et al. showed that a polygenic risk score (PRS) is associated with BC risk in men with BRCA1/2 pathogenetic variants [10].

#### 4.1.1. Germline Pathogenetic Variants in High-Penetrance Genes

Germline BRCA2 pathogenic variants represent the strongest risk factor for MBC. The lifetime risk for breast cancer in males with the pathogenetic variants of BRCA2 is 8.9%, 80–100 times higher than for the general population [11]. The frequency of BRCA2 pathogenetic variants in MBC varies from 4 to 40%; however, the lack of studies with large samples makes a precise estimate impossible. Available trials are retrospective studies with small sample sizes in which the genetic screening methods have varying sensitivities, and missense variants that were not always classified as pathogenic are considered [12].

Germline BRCA1 pathogenetic variants are less frequent, accounting for up to 4% of MBC. The lifetime risk for breast cancer in the BRCA1 male carrier is just over 1% [13].

Other moderate-/low-penetrance genes seem to be implicated in MBC. Pathogenic PALB2 variants confer an estimated 5.3-fold increase in breast cancer risk in women (95% CI: 1.8–13.2). The risk is high for truncating alterations in PALB2 but not for missense variants [14]. In a recent study, 115 MBC cases, negative for BRCA2 pathogenetic variants, were evaluated for PALB2. PALB2 pathogenetic variants accounted for 1–2% of cases. Data from this study suggest that family history is not a strong predictor of being pathogenetic variant carrier in males, and that MBC patients should be tested for BRCA2, and possibly for PALB2 [12]. PALB2 could be tested in families with recurring breast cancer or pancreatic cancers or in families with aggregation of MBC [15]. The genetic alterations seem to frequently occur in exon 4 and 5 as described in a large population-based study ic Central Italy and in a small series [16,17].

Germline pathogenetic variants in the CHEK2 gene were initially identified in families with Li–Fraumeni-like syndrome but without the characteristic pathogenetic variants in TP53 [18]. Heterozygous CHEK2*1100delC is associated with an increased risk of BC: 2.7-fold for unselected BC, 2.6-fold for early onset BC, and 4.8-fold for familial BC in women. Pathogenic variants in CHEK2 are not associated with an increased risk of sarcoma, such as for TP53 alterations in Li–Fraumeni syndrome [19,20].

Table 1 summarizes the genes involved in the main hereditary breast cancer syndromes and the typical cancer spectrum related to them, including the percentage of risk of breast cancer associated with the predisposing genes in males and females.

#### 4.1.2. Low-Penetrance Alleles and Polygenic Risk Scores

High-penetrance (BRCA1/2) or moderate-penetrance genes (CHEK2, PALB2) cannot entirely explain the genetic susceptibility to MBCs. Recently, single nucleotide polymorphisms (SNPs) in low-penetrance genes, such as ESR1, TOX3, and FGFR2, have been shown to modify the risk of developing breast cancer in a large series of 413 Italian MBCs [21].

Recently, Maguire et al. performed genome-wide single nucleotide polymorphism genetic testing of MBC subjects with European ancestry (1380 male breast cancer). Genetic comparison analysis showed a high and strong common genetic basis between the analyzed MBC cases and estrogen receptor positive FBC [22]. This was further confirmed by Lecarpentier et al. [10]. They observed a sex-specific association between BC and some SNPs. In particular, while SNP rs9371545 and rs11571833 are associated with ER-negative MBC, other SNPs (rs11249433, rs34005590, and rs2981578), which are known as predisposing for ER-positive FBC, are not associated with MBC. These sex-specific differences may be related to either influence of different endogenous factor on the expression of SNPs or to a different activity of the target genes of SNPs in the develomìpment process of male vs female BC [10].

Germline mutations in the androgen receptor (AR) gene have been associated with increased risk of MBC [23]. Primary or metastatic MBC express ARs in about 70–90% of cases. In MBC, the AR gene has a wide sequence of repeats (CAG sequence) that is a highly polymorphic region of glutamine repeats [24].

#### 4.1.3. Somatic Variants

Molecular studies have generally focused on germline pathogenetic variants, with very few studies focusing on somatic changes in MBC. Deb et al. analyzed a small series of familial MBCs for pathogenic variants at somatic level and copy number variations. They found that in the overall population PIK3CA gene alterations are more frequent than TP53 and PTEN alterations. In addition, they found a differential pattern of alterations between BRCA2 and BRCAX tumors, with TP53/PTEN alterations, gain of STK11 SMARCB1, and HRAS, loss of RB1 being more frequent in the first and PIK3CA alterations being more frequent in the latter, suggesting different tumor pathways for BRCA2 and BRCAX MBCs [25]. Rizzolo et al. evaluated 103 samples of primary MBCs, identifying copy number variations (CNVs) in different targetable oncogenes [26]. In particular, ESR1 deletion, EGFR amplification, and CCND1 amplification were found at a higher frequency in BRCA1/2-negative MBC compared to FBC [26,27]. ESR1 deletion was reported as CNV gender specific [27], and it was generally associated with an ER-negative status; however, it was also observed in a small percentage of ER-positive MBC. The identification of ESR1 deletion plays a key role in the therapeutic approach of the latter group, with, in fact, the ESR1 deletion being associated with tamoxifen resistance [26].

EGFR and CCND1 amplification were reported to be independent prognostic factors, identifying two different subsets of MBC, both characterized by poor prognosis, HER2 positivity, and resistance to endocrine therapy [26,27]. These findings may acquire great importance in MBC, leading to a possible earlier use of combined HER2 inhibitors with EGFR inhibitors, and CDK4/6 inhibitors, in the treatment algorithm of MBC with EGFR amplification and CCDN1 amplification, respectively. If confirmed in a genome profiling study, these somatic oncogenic molecular alterations could represent actionable therapeutic targets.

In more recent large studies, variants in BRIP1 (BRCA1-interacting protein 1) were not associated with a relevant increase in BC risk [28,29]. Previously, BRIP1 was included among moderate-penetrance BC susceptibility genes because deleterious variants in BRIP1 were seen in about 1% of BC without BRCA1/2-variants or familial/early onset BCs [30,31]. However, in an Italian study on 126 MBC, 97 cases were selected and tested for BRIP1 mutations. All of them revealed no variants for the BRCA1, BRCA2, CHEK2, and PALB2 pathogenetic variants. A total of five germline sequence alterations in BRIP1 were detected, without a statistically significant difference in the frequency compared to the control population [32].

#### 4.1.4. Epigenetic Factors

Other than germline and somatic BRCA1/2 pathogenetic variants, epigenetic factors that occur at the somatic level, such as hypermethylation or copy number variations, have been associated with MBC and seem to influence the phenotype and probably different outcomes in MBC compared to FBC [33]. Hypermethylation in different tumor suppressor genes can explain gene silencing in breast cancer and could also explain the different pattern of response in a specific subset of MBC compared to FBC and to other MBC subgroups [27,34,35,36]. Deb S. et al. reported commonpromoter hypermethylation (≥30%) in GSTP1, RASSF1A, MAL, TWIST, RUNX3, and RARβ, identifying a relationship with clinical and pathological features [36]. Johansson et al. observed a correlation with worse prognosis in a subgroup of MBC with hypermethylation, identifying methylation levels in the tissue, serum, or plasma as putative biomarkers of the response [34].

### 4.2. Other Risk Factors

Other genetic risk factors may be responsible of cancer development in males. Modifications in the estrogen-testosterone levels are one of the risk factors for MBC as suggested by men affected by Klinefelter syndrome [36]. In Klinefelter syndrome, a genetic disease caused by an extra X chromosome, males present testicular dysgenesis, gynecomastia, low testosterone serum level, and high gonadotropins [37,38], and the risk of MBC is 50-fold increased [39].

Men with gynecomastia also have a high risk of BC. It is unknown whether gynecomastia itself or the pathogenic factors for gynecomastia might be the cause of MBC.

Bilateral BCs have been reported in men that receive exogenous estrogens, such as those treated for prostate cancer and male-to-female transgender persons who receive cross-sex hormone therapy [37,38].

Factors that increase estradiol levels, such as cirrhosis of the liver, obesity, or exogenous assumption, can impact MBC risk.

The risk of FBC and MBC is also associated with exposure to therapeutic ionizing radiation and radiation treatment during childhood. [38,39,40].

Among lifestyle factors, obesity increases the risk of MBC and in particular, a body-mass index (BMI) ≥30 increase the risk of MBC of 80%. Conversely, physical activity seems to be a protective factor [9]. Alcohol consumption does not increase the risk of MBC, whereas the role of smoking as predisposing risk factor for BC in male is still controversial [9].

## 5. Management

### 5.1. Imaging

MBC patients usually presents with signs of locally advanced tumor (nipple and/or skin involvement); this is due to the smaller breast size in men, and probably the later manifestation of breast symptoms. MBC is commonly located in the subareolar region while in FBC, it is commonly in the upper-outer quadrant. Malignant calcifications are less frequent in men than in women, and their mammographic features (i.e., scattered and punctuate) would be considered benign in women. Cystic lesions are rare, which is different from what occurs in women, and they should be considered as suspicious of malignancy. Lobular tissue is not commonly present in male breasts and for this reason cysts are uncommon. For instance, in men papillary carcinomas are generally detected as cysts with a complex pattern at US-imaging. MBC can be erroneously diagnosed for asymmetric/unilateral gynecomastia; however, the first is often eccentric to the nipple while the latter is central and concentric to the nipple [41].

### 5.2. Histopathology

The most frequent histology of MBCs is invasive ductal carcinoma, which occurs in 85–95% of patients. Ductal carcinoma in situ is diagnosed in 5–10% of male with BC [1].

Regarding immunohistochemistry phenotypes, MBC seems to be more likely to express hormone receptors than FBC. In small retrospective studies, 82% of the included MBCs were positive for estrogen receptors (ER)s and 75% showed positivity for progestin receptors (PRs) [1]. A recent large retrospective analysis of 489 male patients confirmed these findings [42].

In this study, ERs were evaluable in 419 tumors, with 92% of tumors being positive; PR status was assessable in 399 tumors and was positive in 89.2% of cases. The concurrent evaluation of both hormone receptors shows ER+/PR+ = 86%; ER+/PR− = 6%; ER−/PR+ = 3.3%; and ER−/PR− = 4.8%. More than 95% of the tumors showed positivity in at least one hormone receptor. However, tumor subtypes seem to be distributed differently from that seen for women and varies according to race/ethnicity. In fact, as reported by Mc Gregor and colleagues, among 606 patients with different ethnic origins triple-negative or ER-positive/PR-negative tumors are more frequent in non-Hispanic black men for which is also reported a poorer outcome [43].

In the EORTC International Male Breast Cancer Program, 1483 MBCs were retrospectively analyzed over 20 years. For tumors without metastases, ductal invasive carcinomas were reported in 84.8% of cases, with grade 2 in 51.5%; ER-positivity in 99.3%; PR positivity in 81.9%; AR positivity in 96.9% and low expression of Ki67 in 61.1%. The most common subtypes in MBC were the luminal A-like (41.9%) and luminal B-like/HER-2-negative (48.6%), whereas less frequent were the HER-2-positive (8.7%) and triple-negative (0.3%) subtypes [39]. Men aged less than 50 years old had poorer outcomes. In tumors with high ER positivity, with high PR positivity, and high AR positivity, significantly longer overall survival (OS) and recurrent-free survival (RFS) was reported. While HER2 expression, Ki67, IHC subtypes or grade were no associated to OS/RFS [44].

Gargiulo et al. also reported a high prevalence of positive hormonal receptor status (88.4% ER+; 81.4% PR+) in MBC, with HER2-positive and triple-negative tumor prevalence being 26.8% and 7.0%, respectively [5].

HER2 expression has been reported to range between 2% and 27% in MBC. Humphries et al. showed a very low expression of HER2 in MBC, accounting for <10%. In the series by Gargiulo et al., a higher expression was reported, probably attributable to the non-systematic evaluation [1,5,45].

Other biomarkers were recently evaluated in 134 MBCs. Cyclin D1 and bcl-2 are frequently expressed (75% and 77%, respectively) in MBCs. BRST2 and p21 are expressed in 56% and 48% of MBCs, respectively; p53 is expressed only in 15% of these tumors and basal phenotype is uncommon [44]. Furthermore, it has been demonstrated that there is an association between high grade, high mitotic count and HER2 amplification and/or overexpression, high Ki67, p53 accumulation, high p21 expression, low PR expression, and low bcl-2 expression. A decreased 5-years survival were statistically associated to PR negativity and p53 accumulation; they were also independent markers of patient prognosis [46].

Histopathological features seem to be related to the tumor mutational status. An Italian multicenter study evaluating 382 male patients with BCs, including 50 BRCA carriers, reported that MBCs in BRCA2 carriers showed a statistically higher tumor grade, PR negativity, and HER2 positivity. Carriers of BRCA2 pathogenetic variants develop more frequently secondary primary tumors (OR = 11.42, 95% CI 1.79–73.08). BRCA2-related MBCs have a specific phenotype characterized by high-grade tumors, PR-negative status and consequently a more aggressive behavior [47].

MBC is often diagnosed at an advanced stage. T4 disease represented 20–25% of cancers [48]. The pT3–T4 stage significantly increases according to age, with the highest percentage (42%) in men over 70 years old [48]. Axillary nodal involvement is present in approximately 50% of MBC [42,48,49] and significantly correlates with the pathological tumor size [42].

### 5.3. Treatments

Treatment options, schedules, and duration of therapies in MBC for both localized and metastatic disease are usually derived from recommendations and guidelines for FBC. Treatment includes the integration of surgery, radiotherapy, and systemic therapies.

Modified radical mastectomy is the preferred surgical approach for MBCs, used in approximately 70% of patients. Less favored approaches are represented by radical mastectomy, especially in older patients, total mastectomy, and lumpectomy with or without radiation [48]. Sarmiento et al. reported, in a large population-based cohort, an evaluation on 16,498 MBCs from the National Cancer Database and showed an improved survival related to treatments, particularly surgery. Increasing age, black race, government insurance, more comorbidities, and higher tumor stages were associated with decreased survival [49]. More recently, Yadav et al. confirmed the association between a worse prognosis and advanced age, black ethnicity, comorbidities, high grade and stages, and poor access to health care [50]. They reported a negative association with mastectomy, differently from the study of Sarmiento. Yadav et al. found that more male patients underwent total mastectomy compared to breast-conserving treatment, which is the preferred option for surgical treatment in females. They showed an association between radical mastectomy and poor clinical outcomes, possibly due to a selection bias because of higher stage associated to larger tumors size and/or node involvement in this cohort. Further evaluation is needed [51].

Moreover, in MBC, like in FBC, sentinel lymph node (SLN) biopsy is a reliable tool for the identification of nodal metastases, as shown by two different experiences in a European and USA Center [52,53].

No strong evidence exists on the use of radiation after mastectomy. Current recommendations suggest adhering to the guidelines for FBC, as shown by a recent single institutional experience [1,54].

Hormonal therapy (HT) represents the gold standard treatment for hormone receptor-positive MBC. Adjuvant HT is represented by tamoxifen for 5 years. Tamoxifen is associated to a decreased risk of recurrence, accounting by 51%, comparable with FBC treatment [55]. In male, about 80% of the circulating estrogen is produced by the aromatase pathway and 20% by the testes [56]. However, the role of adjuvant aromatase inhibitors (AIs) has been little studied in male patients [57]. Eggermann et al., compared adjuvant tamoxifen vs AIs [58].They reported a statistically significant worse outcome in AI cohort for both mortality risk and overall survival [58].

In high-risk patients because of young age, high tumor grade and/or axillary nodal involvement, adjuvant chemotherapy is generally recommended. Currently, anthracycline-based schedules are preferred while in elderly patients, use of the CMF scheme has been described. However, adjuvant chemotherapy conferred a not statistically significant lower time to recurrence and improvement in overall survival [55].

No data exist about the adjuvant use of trastuzumab in MBC, and only one case report speculates about its efficacy in metastatic disease [59]. Considering that there is no biological reason for trastuzumab showing different activity in MBC than FBC patients, this treatment, and pertuzumab, might be considered for HER-2-positive MBC.

Tamoxifen remains the gold standard treatment for male patients. The 5-year OS is similar for tamoxifen-treated FBC and MBC patients (85.1% and 89.2% respectively; *p* = 0.972) [60]. Luteinizing hormone-releasing hormone (LHRH) agonists and orchiectomy are other available therapies. AI associated or not to LHRH agonist was evaluated in a retrospective series, reporting a response rate of 26.1% and a disease control rate of 56.5% [61,62]. In comparison with AI-treated FBC, MBC had a significantly poor outcome (5-year OS was 85.0% in FBC vs. 73.3% in MBC; *p* = 0.028) [60].

HT is indeed associated with a multitude of side effects in men. The most common adverse event related to tamoxifen is hot flashes. Decreased libido, weight gain, and malaise have been described. Less common are rash and erectile dysfunction. Alteration of hepatic function, lung embolism, thrombophlebitis, myalgia, depression, visual deficit, and diarrhea are uncommon [63]. Reported toxicities of AI use include decreased libido, leg swelling, and depression for anastrozole while edema and hot flashes have been reported after use of letrozole [64].

In metastatic MBC, chemotherapy has been offered as second- or third-line therapy after relapse of HT in ER-negative patients. Few and small studies highlight the role of chemotherapy in this setting of patients; the poor efficacy might be related to the old schedules used in those studies [65].

More recently, in the phase III OlympiAD trial germline BRCA1/2 metastatic BC patients were randomized to olaparib, a PARP inhibitor, versus a standard chemotherapy (capecitabine or eribulin, or vinorelbine). The study demonstrated a statistically significant advantage in median progression-free survival (mPFS) in patients treated with olaparib (mPFS = 7.0 months) vs patient treated with chemotherapy (mPFS = 4.2 months). Of note, in this study, about 2% of the enrolled population, respectively 5 out 205 patient in the olaparib group and 2 out of 97 in the contol group were male [66].

## 6. Cancer Genetic Counseling

In the clinical setting, professionals involved in the diagnosis and treatment of MBC patients should refer them to cancer genetic counseling services. Cancer genetic counseling should foresee risk assessment, adequate genetic testing, and an appropriate oncological preventive program for MBC patients and their at-risk healthy family members according to the specific genetic predisposition [67,68].

### 6.1. Risk Assessment

Genetic testing should be offered at diagnosis of breast cancer in a male patient, independently of a positive oncological family history [69]. However, the assessment of the ‘a priori’ probability of identifying a pathogenetic variant is an important component of pre-test counseling. Different probabilistic tools are available for risk assessment. Among these, one of the most widely validated is BRCAPRO [70,71]. Several studies have validated this model relative to FBC [72,73], but few data exist about MBC. In a recent Italian trial, Different prediction models were evaluated for their performance in 102 MBC patients with mutations in BRCA1/2 genes, diagnosed between 1991 and 2007. Thirty-nine out 102 patients (38%) of the patients reported a breast and ovarian cancer diagnosis in first- and/or second-degree relatives. Family history (FH) seems to be a better predictor of pathogenetic variants. Thus, pathogenetic variant carriers account for 15.4% (6/39) of MBC patients with a positive FH and 6.4% (4/63) of those without. Comparing different risk estimation models, only the BRCAPRO version 5.0 showed the best performance with a good sensitivity, specificity, and negative and positive predictive value [74]. More recently, BRCAPRO version 6.0 has been specifically validated as a counseling tool, showing a high performance in determining the ‘a priori’ risk of having a pathogenic variant of MBCs with or without positive FH [75].

The BOADICEA (Breast and Ovarian Analysis of Disease Incidence and Carrier Estimation Algorithm) program is an upcoming tool in European countries [76]. Since 2008, it has been updated and extended, considering the risks of MBC [77]. Previously, two studies compared BOADICEA, BRCAPRO, and other risk estimation models relative to MBC. However, they considered both FBC and MBC, the number of MBC cases evaluated was scarce, and inconsistent results were shown [78,79,80]. More recently, a comparison of BOADICEA, BRCAPRO, and the Myriad probabilistic models was performed in 307 male patients with BC and tested for BRCA1/2. Fifty-eight of these patients were BRCA1/2 carriers. BOADICEA was effective in predicting the total number of BRCA1/2 carriers, and there were non-significant differences in carriers-prediction performance between the BRCAPRO and the BOADICEA. Conversely, Myriad underestimated the number of carriers in almost 70% of the cases and therefore has not been considered a good carriers-predictive model for MBC patients [81].

### 6.2. Genetic Testing

Genetic testing should be offered within pre- and post-test counseling sessions according to the implications of the genetic test itself and the results [67,82,83]. Genetic testing for BRCA1/2 and PALB2 should be the first choice for mutational analysis. BRCA2 and PALB2 own similar clinical phenotype, i.e., a higher risk of solid tumors in children carriers of homozygotes pathogenetic variants. In addition, there is also an increase risk of developping BC in women. BRCA1/2 and PALB2 pathogenetic carriers in heterozygosis have also higher relative risk for pancreatic cancer [84,85,86]. BRCA2 and PALB2 share an analogue function and cancer predisposing spectrum; therefore, PALB2 genetic testing should be proposed to MBC patients belonging to families with BRCA1/2-negative or no informative genetic test results and to families in which a typical tumor of the BRCA2 spectrum are diagnosed. Other genes, i.e., TP53 and PTEN, should be considered if the family history is suggestive of minor syndromes, such as Li–Fraumeni and Cowden, respectively [69]. Moreover, mutational analysis of moderate-/low-penetrance genes should be considered if the genetic test is negative or not informative of pathogenetic variants in BRCA1/2. MBC patients should be counseled for the potential hereditary risk for relatives. Carriers of BRCA2 pathogenetic variants should be advised on the risk of Fanconi anemia and/or brain tumors in minor children if BRCA2 variants has been revealed in both maternal and paternal lines [87,88,89]. The most common pathogenetic variant situation would be in Iceland or in Ashkenazi Jews and no reports of Fanconi in double hits in Iceland or of c.5946delT have been described.

Figure 1 summarizes the genetic testing process for MBC.

### 6.3. Management of Male Carriers of BRCA1/2 Pathogenetic Variants

BRCA status can impact on preventive options according to the cancer spectrum related to BRCA1 or BRCA2 pathogenetic variants. Moreover, the differential diagnosis of hereditary breast and/or ovarian cancer (HBOC) syndrome related to BRCA mutations and other minor syndromes, i.e., Li–Fraumeni and Cowden syndrome, on clinical basis and molecular definition by genetic testing can provide a new perspective in terms of prevention [69]. MBC patients have an increased risk of developing second primary tumors. In the SEER database review, that includes 4873 MBC cases, it has been reported a 1.9% incidence of second primary malignancy [90]. In this study, 21% of MBC patients received a diagnosis of a different primary tumor, such as prostate, colon, or genitourinary cancer.

BRCA pathogenetic variants are associated with higher risk of prostate cancer [52,91]. By panel testing, pancreatic cancer occurs in 0–3% for BRCA1 and 1–6% for BRCA2 [92,93,94,95].

Recently, data on the differences in cancer risk association in male patients with BRCA1 and BRCA2 pathogenetic variants have emerged. Male BRCA2 carriers are more frequently affected by breast (OR, 5.47; 95% CI, 4.06–7.37; *p* < 0.001) and prostate (OR, 1.39; 95% CI, 1.09–1.78; *p* = 0.008) cancers and pancreatic cancers (OR, 3.00; 95% CI, 1.55–5.81; *p* = 0.001). Male BRCA2 carriers show an increased risk for developing two (OR, 7.97; 95% CI, 5.47–11.60; *p* < 0.001) and three (OR, 19.60; 95% CI, 4.64–82.89; *p* < 0.001) primary tumors [96,97]. In addition, Barnes et al. applied polygenic risk scores (PRS); which match specific SNPs associated to various diseases, to further stratify the class of risk in BRCA1/2 carrier patients. Silvestri et al. used the specific SNP breast and prostate cancer PRS to stratify male BRCA1 and BRCA2 carrier patients based on their absolute risk of developing BC and PC. This ultimately permitted the identification of two subgroups of male carriers, with the first having a higher risk of BC and PC and the second having low risk [98]. Taken together, all these results and their further validation could allow, in the near, future optimization of the screening surveillance of male BRCA1/2 carrier patients, offering an intensified and anticipated screening for high-risk carriers, and limiting and postponing prevention for low-risk carriers.

However, currently, the National Comprehensive Cancer Network (NCCN) recommends the recruitment of all males carrying BRCA1 or BRCA2 pathogenetic variants within the same screening programs, especially for breast and prostate cancer [99]. More recently, the NCCN included melanoma in its screening algorithm in this subset [69] and recommend mammography only in males with gynecomastia or parenchymal/glandular breasts density [100].

In Table 2, screening strategies are summarized according to the diagnosed hereditary cancer syndromes and NCCN guideline [69].

## 7. Conclusions

MBC has different pathogenic factors compared to FBC. Genetic alterations are consistent risk factors of the pathogenesis of MBC. Cancer genetic counseling should be included in the diagnostic work-up of MBC patients to improve preventive options while also considering their susceptibility to cancers at different sites. Current treatment of MBC is based on studies conducted in the female setting; however, MBC seems to somehow be associated with different molecular and biological drivers compared to FBC. Therefore, further studies, specifically designed for MBC, are needed to clarify the optimal management. Finally, recent evidence on the epigenetic changes that occur at the somatic level could lead to future tailored therapies in specific subsets of MBC.

## Figures and Tables

**Figure 1 cancers-14-02006-f001:**
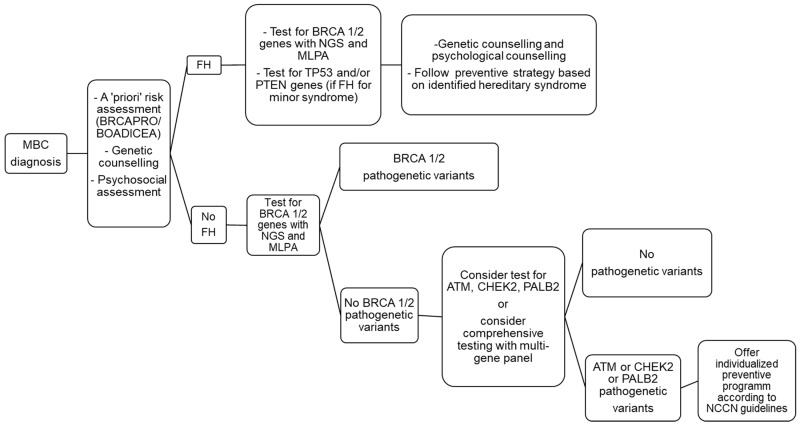
Genetic testing flow chart after a diagnosis of breast cancer (BC) in a male patient.

**Table 1 cancers-14-02006-t001:** Male breast cancer (MBC) risk associated with BRCA1/2 and other moderate-/low-penetrance genes.

Gene	Chromosome	Trasmission	Syndrome	FBC Risk * (%)	MBC Risk (%)	Cancer Spectrum	Contribution to Hereditary Breast Cancer Syndrome
High penetrance							
BRCA1	17q21	AD	HBOC	39–87	1–5%	ovary, prostate, colon, pancreas	20–40%
BRCA2	13q12-13	AD	HBOC	26–91	5–10%	ovary, prostate, pancreas, ductal-gall tract, melanoma	10–30%
TP53	17p13	AD	Li-Fraumeni	56–90	NA	Soft-tissue sarcoma, osteosarcoma leukemia, brain, adrenocortical gland, colon	<1%
PTEN	10q23	AD	Cowden	25–50	NA	thyroid, endometrium, genital-urinary tract	<1%
STK11	19	AD	Peutz-Jeghers	45–54	NA	colon-rectum, small bowel, pancreas, uterus, testis, ovary	-
Moderate-low penetrance							
ATM	11q22-23	AR	Atassia-Teleangectasia	NA	NA	leukemia, lymphoma	-
CHEK2	22q11	AD	Li-Fraumeni variant	24	10-fold	prostate, colon	-
PALB2	16p22	AD AR	HBOC syndrome Fanconi	33–55	NA	Ovary, pancreas, medulloblastoma, Wilms tumor	-

Abbreviations: AD = Autosomal Dominant; AR = Autosomal Recessive; HBOC = Hereditary Breast and/or Ovarian Cancer; FBC = female breast cancer; MBC = male breast cancer; NA = not available, * by age 70 years.

**Table 2 cancers-14-02006-t002:** Preventive strategies for the management of men at risk of hereditary breast cancer syndromes according to National Comprehensive Cancer Network (NCCN)—NCCN version 1.2022.

**HBOC syndrome:** **for carriers of BRCA1/2 pathogenetic variants, for not tested subjects or for subjects belonging to family with an identified BRCA1/2 pathogenetic variant**
Breast self-exam training and education starting at age 35 years Clinical breast exam, every 12 months, starting at age 35 years Consider baseline mammogram at age 50 years; annual mammogram if gynecomastia or parenchymal/glandular breast density on baseline study Consider prostate cancer screening starting at age 40 years Annual full-body skin examination and minimizing UV exposure annual contrast-enhanced MRI for pancreatic cancer screening starting from 50 years or individualized based on cancer present in family
**Li-Fraumeni syndrome:** **for TP53 mutation carriers, for not tested subjects or for subjects belonging to family with an identified TP53 pathogenetic variants**
Address limitations of screening for many cancers associated with the syndrome. Because of the remarkable risk of additional primary neoplasm, screening may be considered for cancer survivors with LFS and good prognosis from their prior tumor(s) Annual comprehensive physical exam every 6-12 months with high index of suspicion for rare cancers and second malignancies, including skin and neurological examinations Consider colonoscopy and upper endoscopy every 2-5 years starting no later than 25 years or 5 years before the earliest known colon cancer in the family Dermatologic examination starting at 18 years, every 12 months Additional surveillance based on oncological family history Pediatricians should be apprised of their risk of childhood cancers in affected families Discuss option to participate in novel screening approaches using technologies within clinical trials when possible, such as whole-body MRI, abdominal ultrasound and brain MRI Brain MRI every 12 months
**Cowden syndrome:** **for carriers of PTEN pathogenetic variants, for not tested subjects or for subjects belonging to family with an identified PTEN pathogenetic variant**
Annual comprehensive physical exam starting at age 18 years or 5 years before the youngest age of diagnosis of a cancer in the family, with particular attention to breast and thyroid exam Baseline thyroid ultrasound at age 18 years Consider colonoscopy starting at age 35 years, then every 5-10 years or more frequently if patient is symptomatic or polyps found Consider annually dermatologic exam Education regarding signs and symptoms of cancer

Abbreviations: HBOC = Hereditary Breast and/or Ovarian cancer; LFS = Li-Fraumeni syndrome; RT = radiation therapy; MRI = magnetic resonance imaging.

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
