# Peer review of "Male Breast Cancer: From Molecular Genetics to Clinical Management"

_cancers, 2022, doi:10.3390/cancers14082006_

Round 1

Reviewer 1 Report

Male breast cancer is rare and generally understudied. There have been a plethora of review articles on male breast cancer so at first glance this is just another. However the authors have focused on describing epidemiology, genetic and non-genetic risk factors so it's not just a 'more-of-the-same' review, which sets it apart. As such it provides a useful addition to the field, though a few key references are missing. The following comments will improve the content:

1) Section 5.2 states "...the incidence of HER2 positivity tumors is 
higher in MBC than in FBC." This in incorrect. HER2 is rarely expressed in MBC. Pls refer to (and reference) work by Humphries et al (Sci Rep, 2017) which compared clinicopathological features in MBC studies published since 1996 which examined >30 cases. This clearly shows from these combined studies that HER2 expression is <10%. Pls rewrite accounting for this information. The EORTC International Male Breast Cancer Program studied1483 MBC but HER2 status was not measured specifically. Instead these cases were stratified into molecular subtypes. The study by Gargiulo et al (ref 5) states 27% of cases were HER2+, form a total of studied 47 MBC. However these cases were from 1989 - 2014. HER2 overamplification in breast cancer was discovered in 1987 by Slamon but it took several years before testing was widely implemented so its likely that part of the cohort used by Gargiulo would not be tested skewing the number they found positive.

2) Section 5.3 should include some comments on clinical trials (or otherwise) for MBC.

3) In section 6.2 authors state "Genetic testing for BRCA1/2 
and PALB2 should be the first choice for mutational analysis". A comment on how mainstream this is is required, alongside information on how this is conducted.

4) There aer many abbreviations in Figure 1 (and Figure 1 should start with a capital letter). Pls check these have all been defined in the Fig legend or text.

Minor/typos

1) In section 5.2 fenotypes should read phenotypes

2) In Conclusions a space is needed between MBC and care (this last line)

3) References 10 and 90 are cited inconsistently (probably due to a software issue) as they have all names spelled out in full, whereas the rest use initials. Pls correct.

Author Response

Dear reviewer,

we would like to thank you for the positive feedback and helpful comments for correction or modification of this manuscript, following which we feel the manuscript is further improved.

The manuscript has been revised to address your comments. We have corrected the incorrect data reported in section 5.2. and some sections have been updated with data from more recent trials and papers.

Along with this we have corrected some typos and reported all the abbreviations, included the ones from the figures, at the end of the manuscript.

We very much hope you will enjoy and accept for publication the revised manuscript.

Sincerely,

Matilde Pensabene

Claudia von Arx

Michelino De Laurentiis

Reviewer 2 Report

The authors have done a commendable job of compiling a review on Male breast cancer.MBC is rare and therefore understudied.Efforts are needed focus on the etiology and genetics of MBC.

The authors have well demarcated the review into pertinent sections.Risk factors have been dealt in detail and important information like genetic rsik factors,germline pathogenic variants,low penetrance alleles and polygenic ris scores and somatic events are well detailed and provide updated information.

Management of the disease includes imaging,histopathology and treatments.

Finally, the review discusses the cancer genetic counselling for MBC.

I would recommend the Editor to accept the review for publication.

Author Response

Dear reviewer,

we really appreciated your comments and we are enthusiastic that you enjoyed reading review. 

We have go through the paper and corrected the spelling mistakes as well as we have clarified some sentences that were unclear. 

We wish you will appreciate these modifications, 

Many thanks and kind regards

Reviewer 3 Report

This is a review article about male breast cancer and I read with much interest. It is a very important but under-discussed topic and I commend the authors on their efforts. The article is well-written with some grammatical and spelling errors, and some unclear sentences. I would recommend the authors to go through and proofread once more. I also find the use of articles and references to be slightly biased, with mostly Italian studies mentioned throughout the MS. I would like to see a more balanced review. The articles referenced in this review paper are okay but I would like to see inclusion of more recent studies, for e.g. Yadav et al, Sarmiento et al, Silvestri et al, Reichl et al, etc. Overall, I have enjoyed reading and reviewing the article.

Author Response

Dear reviewer,

Thank you for the positive feedback and helpful comments for correction and modification to our manuscript.

The manuscript has been revised to address your comments and as you suggested more recent data have been added and cited in the manuscript and we feel that this has further added novelty to the manuscript, therefore we thank you once again. 

Along with this we have corrected some typos and paraphrased some sentences to render them more clear.

We very much hope you will enjoy the revised manuscript.

Kind regards,

The authors